# Determinants of COVID-19 Vaccine Hesitancy in Portuguese-Speaking Countries: A Structural Equations Modeling Approach

**DOI:** 10.3390/vaccines9101167

**Published:** 2021-10-12

**Authors:** Álvaro Francisco Lopes de Sousa, Jules Ramon Brito Teixeira, Iracema Lua, Fernanda de Oliveira Souza, Andrêa Jacqueline Fortes Ferreira, Guilherme Schneider, Herica Emilia Félix de Carvalho, Layze Braz de Oliveira, Shirley Verônica Melo Almeida Lima, Anderson Reis de Sousa, Telma Maria Evangelista de Araújo, Emerson Lucas Silva Camargo, Mônica Oliveira Batista Oriá, Isabel Craveiro, Tânia Maria de Araújo, Isabel Amélia Costa Mendes, Carla Arena Aparecida Ventura, Isabel Sousa, Rodrigo Mota de Oliveira, Manuel Simão, Inês Fronteira

**Affiliations:** 1Global Health and Tropical Medicine (GHTM), Institute of Hygiene and Tropical Medicine, 1349-008 Lisbon, Portugal; 2Epidemiology Center, Feira de Santana State University, Feira de Santana 44036-900, Brazil; julesramon@gmail.com (J.R.B.T.); araujo.tania@uefs.br (T.M.d.A.); 3Institute of Collective Health, Federal University of Bahia, Salvador 40170-110, Brazil; iracemalua.enfermeira@gmail.com; 4Health, Education and Work Department, Federal University of Recôncavo da Bahia, Santo Antônio de Jesus 44380-000, Brazil; fernandasouza@ufrb.edu.br; 5Data and Knowledge Integration Center for Health—CIDACS, Salvador 41745-715, Brazil; andreaferreiracv@gmail.com; 6Human Exposome and Infectious Diseases Network (HEID), University of São Paulo, Ribeirão Preto 14040-902, Brazil; guilherme.schneider@usp.br (G.S.); hericacarvalho@usp.br (H.E.F.d.C.); layzebraz@usp.br (L.B.d.O.); lucmrg0@gmail.com (E.L.S.C.); 7Center for Research in Collective Health, Federal University of Sergipe, São Cristóvão 49100-000, Brazil; shirleymelo.lima@gmail.com; 8Study Group on Health Care, Federal University of Bahia, Salvador 40110-909, Brazil; anderson.sousa@ufba.br; 9Nursing Department, Federal University of Piauí, Teresina 64049-550, Brazil; telmaevangelista@gmail.com; 10Nursing Department, Federal University of Ceará, Fortaleza 60020-181, Brazil; profmonicaoria@gmail.com; 11Global Health and Tropical Medicine (GHTM), Institute of Hygiene and Tropical Medicine, Universidade Nova de Lisboa, 1349-008 Lisbon, Portugal; isabel.mr.craveiro@gmail.com (I.C.); ifronteira@ihmt.unl.pt (I.F.); 12Ribeirão Preto College of Nursing, University of São Paulo, Ribeirão Preto 14040-902, Brazil; iamendes@usp.br (I.A.C.M.); caventura@eerp.usp.br (C.A.A.V.); rodrigoo@usp.br (R.M.d.O.); 13Ministério da Saúde de São Tomé e Príncipe, Cidade Capital, Sao Tome and Principe; a21001261@ihmt.unl.pt; 14Higher Institute of Health Sciences, Agostinho Neto University, Luanda 56910-999, Angola; msimao40@hotmail.com

**Keywords:** vaccines, COVID-19, SARS-CoV-2, vaccine hesitancy, global health, disinformation, infodemic

## Abstract

COVID-19 vaccine hesitancy (VH) has caused concerns due to the possible fluctuations that may occur directly impacting the control of the pandemic. In this study, we aimed to estimate the prevalence and factors associated with COVID-19 VH in Portuguese-speaking countries. We developed a web survey (N:6,843) using an online, structured, and validated questionnaire. We used Measurement Models, Exploratory Factor Analysis, Exploratory Structural Equation Models, and Confirmatory Factor Analysis for the data analysis. The overall prevalence of COVID-19 VH in Portuguese-speaking countries was 21.1%. showed a statistically significant direct effect for VH: vaccine-related conspiracy beliefs (VB) (β = 0.886), perceived stress (PS) (β = 0.313), COVID-19 Misinformation (MIS) (β = 0.259) and individual responses to COVID-19 (CIR) (β = −0.122). The effect of MIS and CIR for VH was greater among men and of PS and VB among women; the effect of PS was greater among the youngest and of VB and CIR among the oldest. No discrepant differences were identified in the analyzed education strata. In conclusion, we found that conspiracy beliefs related to the vaccine strongly influence the decision to hesitate (not to take or to delay the vaccine). Specific characteristics related to gender, age group, social and cognitive vulnerabilities, added to the knowledge acquired, poorly substantiated and/or misrepresented about the COVID-19 vaccine, need to be considered in the planning of vaccination campaigns. It is necessary to respond in a timely, fast, and accurate manner to the challenges posed by vaccine hesitancy.

## 1. Introduction

Vaccination is currently the main public health strategy at a global level, and despite the challenges, it has shown a potential to curb the spread of SARS-CoV-2 infection and its more severe impacts such as hospitalization and deaths [1], being thus of great collective interest. When a certain percentage of individuals is immunized, collective immunity is achieved and, consequently, mitigation and pandemic suppression [2]. However, COVID-19 vaccine hesitancy (VH), that is, the refusal of vaccines, deliberate delay, or incompleteness of vaccine schedules, despite the proven safety, efficacy, and availability of vaccines in healthcare services [3], has worried authorities due to the possible fluctuations that may occur, directly impacting the control of the pandemic [4,5].

The determining reasons for VH are complex and diverse in form and intensity, according to time, location, epidemiological scenario, type of vaccine, and target audience [6,7]. Misinformation and the massive spread of false information, conspiracy theories, and rumors about COVID-19 vaccines, in addition to political polarizations, are among the main factors that have been worrying countries, even before the beginning of vaccination, and have directly and indirectly strengthened VH in different populations, contributing to delayed reach of collective immunity [8,9]. Studies have highlighted high rates of COVID-19 VH ranging between 20 and 50% in countries in the Middle East, Africa, Europe [5] and Latin America and the Caribbean [10,11].

Understanding the VH phenomenon can enable more adequate strategies for its reduction and increased vaccine acceptance. Although it is of paramount importance to know the specificities and characteristics associated with the groups most resistant to vaccination within each country, from a global health perspective, investigating countries and cultures together can provide interesting findings that allow for actionable public health interventions. In this sense, so far, the literature on VH in Portuguese-speaking countries is incipient. Thus, the present investigation aimed to estimate the prevalence and factors associated with COVID-19 VH in Portuguese-speaking countries.

## 2. Materials and Methods

### 2.1. Study Design and Location

An observational, analytical study was conducted through online data collection (web survey), with individuals from seven countries whose official language is Portuguese (Angola, Brazil, Cape Verde, Guinea-Bissau, Mozambique, Portugal, and São Tomé and Príncipe) from May to August 2020.

### 2.2. Sample

The sample size was calculated in the G Power software (version 3.1.9.7) (Düsseldorf, Alemanha), considering the total population size of the countries of interest (*n* = 286,165,991), an incidence rate of the studied phenomenon of 50% (because there are no previous studies with this population); a tolerable error of 3%, a sample design effect correction of 2, a 95% confidence level, and an additional number of 20% participants to compensate losses and refusals. The minimum sample size was calculated at 2562 participants.

A snowball sampling procedure was employed and adapted to the virtual environment in two stages: (I) initially, 30 people (seeds) from Brazil and Portugal were randomly selected from a database of previous studies, ensuring diversity of location in the countries (regions), origin (native or immigrant), race/color (white and non-white), age (young, adult, and older adult) and education (elementary/high school, university education, and graduate education); (II) then, each participant was asked to recruit other individuals in the same category as theirs through their digital social networks.

Additionally, we used mailing lists from university institutions (Institute of Hygiene and Tropical Medicine, University of São Paulo, Angola Higher Institute of Health Sciences, and Mozambique Higher Institute of Science and Technology) and Facebook^®^ publications.

A total of 7,083 people answered the questionnaire. Portuguese speakers residing in others countries (*n* = 127), under 18 years (*n* = 21) who failed to complete more than 50% of the mandatory questions (92) were considered ineligible. Thus, the final sample consisted of 6843 participants.

### 2.3. Data Collection Instruments

A structured online questionnaire was used, developed by the authors based on the scientific literature, and evaluated by experts for cultural and linguistic adequacy and construct validation. In this study, the following blocks of questions were included in the analysis:

1. Sociodemographic data (place of origin; country, region/district, state, and city in which the participant was currently residing; time living in the country; number of people living with the participant; number of rooms in the house; family income; age; gender identity; educational level; religious affiliation, and marital status). 

2. Perception of the COVID-19 pandemic (fear, impact, and limitations experienced during the pandemic, agreement with measures to prevent and mitigate the coronavirus); practices to prevent contact and spread of the coronavirus; and COVID-19 testing and results.

3. Search and consumption of information/news about COVID-19; impact resulting from getting COVID-19 information/news; actions taken based on information/news about COVID-19.

4. Agreement with content related to COVID-19 misinformation, assessed on a Likert scale ranging from “strongly agree”, “agree”, “neutral”, “disagree”, and “strongly disagree”.

5. Willingness to get vaccinated (in case of refusal, the determining reason for the hesitation was asked) and conspiratorial beliefs about COVID-19 vaccines.

The outcome variable was VH, assessed by the question: “Will you get the COVID-19 vaccine when it is available to you?”, evaluated on a dichotomous scale (“yes”/”no”).

### 2.4. Conceptual Structure and Study Hypotheses

A Directed Acyclic Graph (DAG) was constructed [12,13] to represent the conceptual structure and the study hypotheses, in which the directly observed variables are represented by rectangles and latent variables by circles (Figure 1).

The use of DAG has been widely encouraged in the field of causal investigation in epidemiology [12,14,15], as it allows for a clear codification and explanation of the conceptual hypotheses that will be evaluated [13]. In the DAG framework, the causal hypotheses are represented by vertices and edges: the vertices represent the variables, and the edges show a relationship between a pair of variables, signaling the direct or indirect causal paths [13]. Direct causal paths are represented by arrows connecting one variable to another; indirect ones are represented by a sequence of arrows that pass through intermediate variable(s)/mediator(s) of the relationship between two other variables [12]. The latent constructs and the observed variables that made up the structure of the theoretical model are defined below.

COVID-19 misinformation (MIS): this is a serious threat to public health and international relations [16], defined as misinformation caused by incorrect information [16,17] related to the COVID-19 pandemic, whether disseminated knowingly or not about the veracity of the facts [17,18], with or without any purpose of intentionally causing harm to someone [17,18]. It ranges from the dissemination of incorrect general beliefs (GB) related to COVID-19, such as harmful health advice, advice on the ingestion of medicines or herbal medicines without scientifically proven efficacy, politically motivated conspiratorial beliefs (CB) about the origin of the virus, the ineffectiveness of prevention and control measures (such as social isolation), and theories on genetic manipulation of SARS-CoV-2 to cause other diseases [16,17]. COVID-19 misinformation increases fear, causes suffering, produces high levels of stress, intensifies social conflicts, and causes direct damage to people’s health, with social networks being important resources of this content [19]. MIS was treated as a latent construct in the analyses.

COVID-19 individual responses (CIR): this construct refers to COVID-19 coping strategies adopted by people daily. Individual responses depend on people’s perception of risk [20] and are strongly influenced by the type of information received [16], whether from close family and peers, social networks, or official government agencies. It is noteworthy that MIS can generate disastrous responses at the individual level, resulting in increased spread of the disease, increased severity of cases, and even early deaths. Thus, individual responses guided by quality information are crucial to the success of the global responses to the COVID-19 health crisis [19]. CIR was considered as a latent construct in the analyses.

COVID-19 suffering (COVS): The COVID-19 pandemic, like other major global health crises, has a high potential to cause human suffering. This suffering may result from the risk experienced by people in their daily lives of being infected by SARS-CoV-2, which causes anguish, fear, anxiety, and stress [21,22], or even from preventive measures, such as social isolation. These feelings are exacerbated by the possibility of transmission to other people [21,22], hospitalization [22], and close contact with people who had COVID-19 or died from the virus [23,24]. The COVS indicator was defined by summing the affirmative answers (0 = no; 1 = yes) to the following items: 1. Fear of repercussions in life; 2. Impact of social distancing in life; 3. Testing positive for COVID-19; 4. Hospitalization by COVID-19; 5. Close contact with a person who had COVID-19; and 6. Close contact with a person who died due to COVID-19.

Vaccine conspiracy beliefs (VB): conspiracy beliefs about vaccines are widely endorsed by the general population [25] and represent a major obstacle to the successful control of COVID-19 [7,26]. These beliefs accentuate fear and mistrust about vaccines [25] and, combined with misinformation [6,26], make people decide against vaccination [27]. This construct involves ideas intended to diminish trust in vaccines, governments, healthcare professionals, and the pharmaceutical industry [28], such as the notion that pharmaceutical companies and governments falsify vaccine data to their benefit [29] and falsify data on the effectiveness of vaccines or omit evidence about adverse reactions [30]. These conspiracy theories are widespread on social networks [26] and contribute to VH [31]. In this study, the VB indicator was defined by rating scores (0 = Do not believe the news; 1 = Believe the news are true) for the following items: 1. The immunity conferred by vaccines against COVID-19 is short-lived; 2. COVID-19 vaccines alter DNA; 3. The vaccine can cause other diseases such as autism or autoimmune diseases; 4. COVID-19’s vaccine contains chips implanted to control people; 5. The vaccine’s efficacy and the published studies are false. After calculating the score, the indicator was categorized according to the number of conspiracy ideas people believed, as follows: 0 = none; 1 = one news; 2 = two news; 3 = three news; 4 = four news or more. 

Perceived stress (PS): high levels of perceived stress associated with COVID-19 have been shown. Some factors are associated with this phenomenon, including the ineffective strategies developed by health authorities to the detriment of scientific recommendations [32]. In this study, the presence of perceived stress was investigated in a self-reported way (0 = no; 1 = yes) through the following question: Did you feel stressed during the COVID-19 pandemic?

### 2.5. Data Analysis Procedures

Data were organized in SPSS version 24.0 for analysis. Then, the hypothesis testing was conducted within a Structural Equation Modeling (SEM) framework using Mplus software, version 8.4. Exploratory Factor Analysis (EFA) and Exploratory Structural Equation Models (ESEM) were estimated to assess the factor structure, followed by Confirmatory Factor Analysis (CFA) to validate the dimensionality of the measures [33,34]. The criteria for loading the items were: standardized factor loading ≥ 0.3 and residual variance ≤ 0.7 [34,35,36].

Then, the structural model was designed, composed of measured latent variables and directly observed variables. For this purpose, the crude and standardized regression coefficients were estimated with a significance of 5%, and the direct, indirect, and total effects were classified as: weak/small (around 0.10), moderate/medium (close to 0, 30) and strong/large (>0.50) [36]. All analyses were performed in the general population and by gender, age, and education strata, using the weighted least squares means and variance adjusted (WLSMV) estimation. Age strata were defined to enable the comparison between groups of younger, middle-aged and older adults considering the age of the sample studied. The variable level of education was dichotomized into Elementary/High School (9 years of age or less which corresponds to less education) and university education (higher education). These adjustments were necessary due to the differences observed between countries. For re-specification of the model, the modification indices (MI ≥ 10) and the expected parameter changes (MEP ≥ 0.25) [37] were evaluated. To assess the fit of the models, the root means square error of approximation (RMSEA < 0.06—exceptionally < 0.08, with a 90% confidence interval less than 0.08) [37,38], the comparative fit index (CFI ≥ 0.95), and the Tucker-Lewis index (TLI ≥ 0.95) [36] were adopted.

### 2.6. Ethical and Legal Aspects

An Institutional Review Board approved the research project. All participants signed an informed consent form prior to entering the study.

## 3. Results

### 3.1. Sociodemographic Data

In total, 6843 people participated in the study, with a predominance of women (70.4%), aged between 30 and 49 years (48.2%), with university education (79.2%). Most participants agreed with the government’s strategies against the pandemic (68.6%), self-reported perceiving stress (86.0%), and affirmed not having used early treatments for COVID-19 (78.2%). Regarding the experience of the pandemic, the main reports included: fear of the repercussions of the pandemic in the future life (89.4%), suffering impact from social distancing (54.3%), testing positive for COVID-19 (35.5%), and having close contact with people who had had COVID-19 (52.4%) or died due to COVID-19 (11.3%). Among the conspiratorial beliefs about the COVID-19 vaccine, the belief that “the vaccine can cause other diseases, such as autism or autoimmune diseases” (19.1%) stood out (Appendix A).

### 3.2. Prevalence of Vaccine Hesitancy

The overall prevalence of COVID-19 VH was 21.1%. There was a higher prevalence of VH among women (23.0%; CI95%: 21.8–24.2), older adults (29.9%; CI95%: 27.3–32.6), people with a high level of education (22.4%; CI95%: 21.3–23.5), participants reporting perceived high stress (21.9%; CI95%: 20.9–23.0), participants that have used early treatments in the presence of symptoms (26.3%; CI95%: 13.4–43.1), people afraid of future repercussions of the disease (22.4%; CI95%: 21.3–23.4), participants that have tested positive for COVID-19 (23.3%; CI95%: 17.5–25.7), and people in close contact with someone who had COVID-19 (22.2%; CI95%: 20.1–23.6) or died from the disease (24.6%; CI95%: 21.6–27.8). It is noteworthy that all people (100%) who reported agreeing with the statements “COVID-19 vaccines alter DNA”, “the vaccine can cause other diseases, such as autism or autoimmune diseases” or “the COVID-19 vaccine contains chips implanted to control people” reported being hesitant about the vaccine (Appendix A).

### 3.3. Latent Variable Measurement Models

The measurement of latent constructs was initially conducted by Exploratory Factor Analysis, with satisfactory adjustment indices. The high residual correlation between Conspiracy beliefs (CB) and General beliefs (GB) corroborated the convergence of the second-order factor “COVID-19-misinformation” (MIS), also suggested by the Confirmatory Factor Analysis. The evaluation of the residual correlation between the latent constructs COVID-19 misinformation (MIS) and “COVID-19 individual responses” (CIR) had a high and adequate discriminant validity (r < 0.90) (Appendix A).

The factor loadings of the measurement models were high and statistically significant. As for the latent construct CIR, the highest load was observed for the indicator “hand hygiene with soap and water or alcohol” (R3) (λ = 0.777) and the lowest for “using disinfectants for cleaning the environment” (R2) (λ = 0.511). The MIS measurement model was initially evaluated by the first-order factor indicators CB and GB, both with high and statistically significant factor loadings. As for the CB construct, the highest load was observed for the item “positive asymptomatic people do not transmit the virus to other people” (C4) (λ = 0.871), and the lowest for “social isolation can reduce immunity and facilitate virus infection” (C3) (λ = 0.458). As for GB, the highest load was observed in the item “autohemotherapy is very efficient against the new coronavirus” (G6) (λ = 0.896), and the lowest in the item “alcoholic solution is more efficient than washing hands with soap and water as a preventive measure” (G2) (λ = 0.496). The high correlation between the CB and GB constructs (r = 0.949) endorsed the convergence of the second-order factor, MIS. Residual correlations between items did not indicate the need for re-specification of the model (Table 1).

### 3.4. Stratified Measurement Models

The stratified measurement models presented high standardized factor loadings and satisfactory adjustment indices, except for models 3.1 and 3.2 (education strata), in which the item “hand hygiene with soap and water or alcoholic solution” (R3) of the latent variable “COVID-19 individual responses” (CIR) had a low load (<0.30), suggesting its exclusion. Besides, in the education strata, the variables “vinegar is better than alcohol to avoid contamination by COVID-19” (G5), “eating garlic prevents the new coronavirus” (G7), and “drinking water every 15 minutes expels the new coronavirus, as it prevents it from going to the lungs” (G9) had empty cells in the bivariate proportion comparison, implying the need to collapse response categories. After re-specification, the models stratified by education showed satisfactory fit indices and high and significant loads (data not shown in tables).

### 3.5. Structural Equation Model of SARS-CoV-2 Vaccine Hesitancy

In the structural model, vaccine hesitancy (VH) was considered the response variable, COVID-19 Misinformation (MIS) and COVID-19 individual responses (CIR) the explanatory and latent variables, and COVID-19-related suffering (COVS), vaccine conspiracy beliefs (VB), and perceived stress (PS) were the observed variables. The fit indices of the structural model were satisfactory (Figure 2).

A statistically significant direct effect of the following factors on VH was found: VB (β) = 0.886; *p*-value < 0.001), PS (β) = 0.313; *p*-value < 0.001), MIS (β) = 0.259; *p*-value < 0.001), and CIR (β) = −0.122; *p*-value < 0.001), indicating that conspiratorial beliefs, perceived stress, COVID-19 misinformation, and ineffective individual responses are related to vaccine hesitancy. There was a strong direct effect of VB, a moderate effect of MIS and PS, and a weak effect of CIR on VH. Thus, VB was the factor that most contributed to VH (Figure 2).

In the evaluation of specific, statistically significant indirect paths, it was identified that: (a) the higher the MIS, the greater the VB and, consequently, the prevalence of VH, with a moderate effect of MIS on VB (β) = 0.364; *p*-value < 0.001); (b) the lower the COVS level, the greater the perceived stress (β) = −0.159; *p*-value < 0.001) and the prevalence of VH; c) the lower the VB, the higher the level of PS and VH, with a moderate effect of VB on PS (β) = −0.300; *p*-value < 0.001); (d) PS level was an important mediator of the relationship between COVS, VB, and VH; (e) VB and COVS contributed to the increase in the level of perceived stress, with emphasis on VB; and (f) the lower the COVS level, the higher the CIR and VH index, with a weak and significant effect of COVS on CIR (β) = −0.080; *p*-value = 0.005) (Figure 2). 

### 3.6. Stratified Structural Equation Models of SARS-CoV-2 Vaccine Hesitancy

The stratified analysis made it possible to compare the effects between subgroups divided by gender, age, and education. COVS remained without a statistically significant direct effect on VH in all subgroups. When divided by gender, a direct effect of MIS (β) = 0.317; *p*-value < 0.001) and CIR (β) = −0.152; *p*-value < 0.001) on VH was found: PS was higher among men (β) = 0.344; *p*-value < 0.001) and BV was higher among women (β) = 0.953; *p*-value < 0.001). The factors that most directly contributed to VH among men were VB (β) = 0.795; *p*-value < 0.001), with a strong effect, and MIS, with a moderate effect, and, among women, VB and PS with a strong and moderate effect, respectively (Table 2).

There was a significant negative relationship of COVS in PS among men (β) = −0.265; *p*-value < 0.001); thus, the greater the suffering related to COVID-19, the lesser the perceived stress. Unlike the general model, among women, there was a statistically significant effect of COVS on VB—the greater the suffering, the greater this type of belief (β) = 0.068; *p*-value < 0.001) and VH. The effect of COVS on CIR (β) = -0.180; *p*-value < 0.001) was significant only among men, with an effect twice as great as that found in the general model. This result suggests that, for men, the less the suffering, the greater the individual response and the lower the VH. There were no discrepancies in the effect of MIS on VB by gender compared to the general model. There was a significant effect of PS on CIR among men (β) = 0.138; *p*-value < 0.001) diverging from the general model, indicating that the greater the perceived stress, the greater the individual response and the lower the VH. The effect of VB on PS was significant among women ((β) = −0.461; *p*-value < 0.001) when compared to men ((β) = −0.163; *p*-value < 0.001); that is, the lower the level of conspiratorial beliefs in the vaccine, the greater the stress, especially among them, and the greater the VH (Table 2).

In models stratified by age, the effect of PS for VH was greater among younger people (β = 0.578; *p*-value < 0.001), the effect of MIS was greater among people aged 30 to 49 years (β = 0.673; *p*-value < 0.001), and the effects of VB (β = 0.906; *p*-value < 0.001) and CIR (β = -0.410; *p*-value < 0.001) were greater among the older participants. The factors that most contributed to VH were VB (β = 0.587; *p*-value < 0.001) and PS among the youngest subjects, MIS and VB (β = 0.484; *p*-value < 0.001) in subjects aged between 30 and 49 years, and VB and CIR among the older participants. All these pathways to VH had strong effects, with the effect of VB standing out (Table 3).

The effect of COVS on PS (β = −0.195; *p*-value < 0.001) was significant only in individuals aged 30 to 49 years, showing that, for this age group, the lower the suffering related to COVID-19, the higher the level of perceived stress and VH, although the direct path from PS to VH (β = 0.111; *p*-value = 0.097) was not statistically significant in this stratum. Concerning the effect of COVS on CIR, there was statistical significance in the subgroup aged 18 to 29 years (β = 0.133; *p*-value = 0.024) and in the subgroup aged 30 to 49 years (β = −0.188; *p*-value < 0.001). In the first group, the greater the suffering, the greater the individual response and the lower the VH. In the second group, the lesser the suffering, the greater the individual response, and the lesser the VH. Diverging from the general model, the effects of MIS on PS and of PS on CIR were significant and moderate in the strata from 30 to 49 years (β = −0.304; *p*-value = 0.026) and in younger participants (β = 0.359; *p*-value < 0.001), indicating that in the intermediate age group, the lower the level of COVID-19 misinformation, the higher the level of stress and VH, while, in younger subjects, the higher the stress level, the greater the individual response and the lower the VH. The effect of MIS on VB was significant in the three strata, but it was much more expressive in the group aged 30 to 49 years (β = 0.617; *p*-value<0.001), showing that the higher the level of COVID-19 misinformation, the greater the vaccine conspiracy belief and the VH. The effect of VB on PS was significant among the younger subjects (β = −0.155; *p*-value = 0.027)—the lower the vaccine conspiracy belief the greater the stress and the greater the VH (Table 2). 

As for the models stratified by education, no outlying differences in direct effects for VH were identified. The factors contributing most to VH in both groups were PS and MIS with moderate and strong magnitudes. The analysis of indirect paths did not resulted in marked differences, nor any divergence compared to the general model (Table 2).

### 3.7. Standardized Total and Indirect Effects of the Structural Equation Model of SARS-CoV-2 Vaccine Hesitancy 

In the general model, the greatest total effects on Vaccine hesitancy (VH) were observed for high Vaccine conspiracy beliefs (VB) (β = 0.795; *p*-value<0.001) and high COVID-19 misinformation (MIS) levels (β = 0.523; *p*-value<0.001), both strong and significant. There were also medium total effects of high PS (β = 0.305; *p*-value<0.001) and small of low COVID-19 individual responses (CIR) (β = −0.122; *p*-value = 0.001).

Comparing the total effects in the models stratified by gender, age, and education, it was observed that: (a) the effect of COVID-19 suffering (COVS) on VH was significant only among men (β = −0.117); (b) the effects of MIS on VH were positive and significant in all models, higher among those aged between 30 and 49 years (β = 0.911), and lower among those aged 50 years or more (β = 0.372); (c) the effects of CIR on VH were negative and significant in all models, with greater amplitude among those aged 50 years or more (β = –0.410) and smaller among those with elementary and high school education (β = −0.106); (d) the effects of PS in VH were all positive, not significant only in the age group of 30 years and over, with the greatest effect among those aged between 18 and 29 years (β = 0.474) and the lowest among those aged 30 to 49 years (β = 0.107); (e) the effects of VB on VH were strong, positive and significant in all models, with the greatest magnitude among those aged 50 years or more (β = 0.881) and the lowest among those aged 30 to 49 years (β = 0.474) (Table 3).

Regarding the specific indirect paths of the general model, there were small and significant effects of higher COVS mediated by lower PS (β = −0.050; *p*-value<0.001) and higher CIR (β = 0.010; *p*-value = 0.023) on VH. These effects were stronger in the male strata and remained in the educational strata. Among women, there was significance in the effect of higher COVS mediated by higher VB (β = 0.065; *p*-value = 0.027) and lower VB and PS (β = −0.011; *p*-value = 0.040) (Table 3).

The effect of MIS mediated by VB for VH was significant in all strata, with the greatest effect among women (β = 0.348; *p*-value < 0.001) and older subjects (β = 0.346; *p*-value < 0.001), with a medium magnitude. High MIS for VH, mediated by low VB and PS, was significant in the gender and education strata only in the younger age strata, with the greatest effects observed among women (β = -0.058; *p*-value < 0.001). The effect of high PS for VH, mediated by low CIR, was significant only among the younger participants (β = −0.104; *p*-value = 0.027). The effect of high VB on VH, mediated by low PS, was not significant only into the age strata of 30 years and over, with a significant effect among women (β = −0.159; *p*-value<0.001) (Table 3).

## 4. Discussion

Although most participants in this study (78.9%) demonstrated the intention to be vaccinated against SARS-CoV-2, the prevalence of vaccine hesitancy (VH) (21.1%) can compromise the efficiency of collective immunization. Women, older adults, people with a high educational level, and people who had lost a family member or a friend due to COVID-19 were more likely to hesitate to be vaccinated. These characteristics differ from studies carried out in developed countries [39,40] and reinforce that, although VH is a global problem, specificities of different population groups must be considered and understood.

The attitude towards vaccines can vary from acceptance to total refusal. Between these two extremes lies vaccine hesitancy, which is, at the same time, the attitude that presents greater possibilities of compromising vaccine coverage, and the one that is most susceptible to change [41]. Data indicate that the determining factors for vaccine hesitancy, regardless of the harm to health, are based on a belief system that has not undergone major changes over time. The engagement of those adherents to anti-vaccination movements remained practically unchanged in the last two centuries. However, their content is constantly renewed, and more recently, they have gained the widespread support of social networks [42,43,44].

Vaccine conspiracy beliefs (VB) were identified as the construct most strongly associated with vaccine hesitancy in the general model and seems to relate, at the same time, to the recent global pattern in which availability and access to information foster in individuals a “false empowerment” and a sense of control over diseases that outweigh the need for vaccines produced by “suspicious laboratories and countries” [45,46,47].

Belief systems have a great influence on populations’ attitudes. In the context of VH to COVID-19’s immunizers, “pleasant” beliefs create affective–informational bubbles of disinformation guided by an algorithmic logic, in which people seek information that reinforces their precepts and alleviates their fears and tensions, even if they are based on error and determine a mistaken response to the problem [48,49]. It may have been because of this that the variables Conspiracy beliefs (CB) and General beliefs (GB) were configured as latent constructs that, together, formed a superior construct aimed at accepting misinformation contents, which may have determined, in this study, the individual response of the subjects and the results regarding vaccine hesitancy.

Distorted knowledge of reality generates the wrong individual responses, transforming individuals’ attitudes and practices into socio-cognitive complications and vulnerabilities. For example, a low level of effective individual responses (CIR) resulted in VH. This fact can be explained by risk perceptions (beliefs about potential harms) as predictors of adult vaccination behavior [50]. In the health area, two dimensions are used to assess risk perceptions: vulnerability perception or likelihood of harm if no action is taken; and perception of severity or consequences if the disease occurs [50,51].

Therefore, individuals who do not perceive SARS-CoV-2 infection as a risk, in addition to not avoiding agglomerations and not using masks or adhering to hygiene practices, tend to be hesitant to take the vaccine. It is evident that the lack of awareness, or inadequate information, added to the dissemination of imprecise knowledge about the safety of vaccination, has contributed to increased VH [52].

Beliefs, whether personal or conspiratorial, have a lot of power over real, everyday facts. Although this type of behavior may be related to the low level of health literacy, in our case, it was the more educated subjects who had the greatest tendency to hesitate. This fact can be explained by preferential access to information, regardless of its type. Moreover, the fact that the study was carried out online may explain this variation, simply because it is biased towards a population group with more access to information (particularly relevant in the African countries studied).

Another hypothesis that can explain the findings is the fact that, although there are many counter-disinformation initiatives aimed at COVID-19, their impact is still small, whether due to the high power of dissemination of disinformation contents or the low capacity of these strategies in entering some specified places.

The position of the governing bodies is also fundamental for vaccination adherence and, therefore, low VH. Some of the countries included in this study had their leaders publicly declare ideas aversive to the COVID-19 vaccine, which may have influenced the study findings. On the other hand, in some of the countries studied, such as Guinea Bissau and Mozambique, there is a history of predatory clinical research that does not favor introducing a new vaccine [53,54,55,56]. Another critical problem for vaccine efficacy is the molecular evolution that SARS-CoV-2 has been undergoing, with the emergence of so-called ‘viral variants’.

It is interesting to note that when we consider the VH model stratified by gender, age, and education, considering the specific indirect pathways of the general model, some isolated effects, despite being small, influenced the studied outcomes, so that important differences occurred.

In the literature, available studies on VH indicate that individual factors such as emotions, values, risk perceptions, knowledge, and beliefs, when stratified by personal characteristics such as gender, age, and religion, result in different weights to VH [3,39,52]. In this study, gender and age played a prominent role on vaccine conspiracy beliefs (VB). For example, the factor that most contributed to VH among men, women, and age groups was VB, confirming the impact of conspiracy beliefs on people’s decision to get the vaccine.

In the gender strata, the effect of high MIS and low COVID-19 Suffering (COVS) among and of high Perceived Stress (PS) and high COVS on VH also stood out. Men were more likely to agree with COVID-19 misinformation contents than women [52]. This finding may be related to the fact that men may have a lower understanding of COVID-19 symptoms than women [52]. Due to the low-risk perception, men are probably less able to identify sources of virus propagation or adopt effective prevention measures, such as vaccination, which are objects of attack from misinformation contents. In our study, the relationship between a low level of individual responses to COVID-19 and high VH among men supports this association.

The correlation between high PS and VH among women is a controversial result, as stress is expected to motivate people to engage more in preventive behaviors [57]. On the other hand, the overload of information concerning the pandemic, whether real and reliable or not, is positively associated with cognitive dissonance [58], resulting in a higher VH, since women weigh more risks at the expense of rewards or maximize potential disadvantages. The association between high PS and VH among women was also observed among younger subjects in our study and needed to be better explored longitudinally, using a validated instrument to capture better the perceived stress phenomenon (with greater sensitivity and specificity).

We found that the lower the suffering related to COVID-19 (COVS) among men, the greater the probability of VH, an inverse relationship observed among women. Some aspects can be evoked to understand this association. Men are culturally taught to hide their emotions to sustain an image of virility, strength, and incorruptible health [59]. Due to this social pressure for affective blunting, some male participants may have suppressed or minimized the expression of suffering related to the pandemic, which may have affected the findings. Among women, it is believed that the intensity of suffering related to COVID-19 impacted the capacity for discernment and decision-making regarding vaccination. Fear and psychological suffering are traditionally associated with adopting preventive behaviors; however, when in excess, especially in people who are unable to deal with these feelings, they can culminate in irrational thinking and an inhibition of effective coping responses [57,60].

A surprising finding has been the relationship between the lowest level of individual responses and the highest VH in the older adult strata. Advanced age is associated with greater adherence to COVID-19 protective behaviors, which can be attributed to the high level of risk perception among these subjects, as they are more susceptible to diseases and health-related injuries and, thus, are inclined to protect themselves [61]. However, this finding requires scrutiny, as the study’s cross-sectional nature limited the exploration of this possible association more accurately.

Reinders and colleagues [62] suggest that trust is the most prevalent reason for not getting vaccinated, specifically “being afraid of the vaccination and its effects”. Higher vaccination rates have been found among people who most realize the seriousness of the disease [62], such as women and older adults. Our findings indicate that, to ensure that confidence in the vaccine is strengthened, effective communication strategies that address safety and efficacy issues are needed.

In any case, objective and assertive approaches are needed to overcome misinformation about vaccines, particularly on social networks. The existence of medical opinions directly refuting imprecise claims disseminated on the internet and in primary healthcare settings can be helpful [63]. Messages are crucial but not sufficient, especially depending on the source. Therefore, bringing vaccine development processes closer to people, from the beginning, may be useful.

Our research has some limitations. As this is a cross-sectional study, establishing causal relationships and the direction of associations can be challenging. The countries studied were at different epidemiological times and with quite varied COVID-19 prevalence and mortality rates. Besides, there are notable differences in sociodemographic status among Internet survey participants, which may have influenced the results. Regarding willingness to take the vaccine, a question was asked when the vaccination was not yet taking place, although epidemiological data already showed that the hesitation rate varied [59,60,61].

## 5. Conclusions

The expanded understanding of factors such as misinformation and individual practice/response affecting the collective immunization of COVID-19, especially in a pandemic condition, requires different response strategies to achieve better health conditions. It is necessary to respond quickly, fast, and accurately to the challenges posed by vaccination hesitation, especially in Portuguese-speaking countries, as the determinants are configured as socio-cognitive vulnerabilities added to acquired, poorly substantiated, and distorted knowledge.

In addition to a health surveillance discussion, the governance systems of these countries must act in an articulated manner, modulating and softening these factors to obtain better results, an active immunization of the population, and to demystify beliefs. Vaccine hesitancy is a threat to global public health, as today’s individual hesitation will certainly be tomorrow’s urgency.

COVID-19 vaccination permeates geopolitical, socioeconomic, and global health interests, industries, and multinational groups, a fact that deserves special attention from government officials to improve access, adherence, qualified information, and knowledge of the population. 

## Figures and Tables

**Figure 1 vaccines-09-01167-f001:**
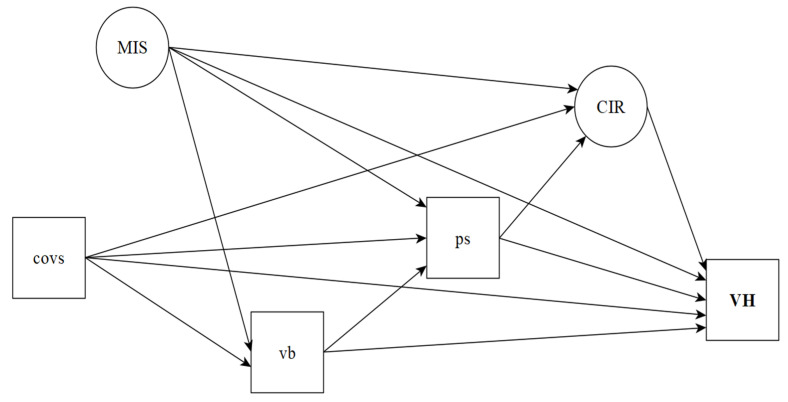
Conceptual structure of determinants of COVID-19 vaccine hesitancy. Notes: COVS: COVID-19 suffering; MIS: COVID-19 misinformation; vb: Vaccine conspiracy beliefs; ps: Perceived stress; CIR: COVID-19 individual responses; VH: Vaccine hesitancy.

**Figure 2 vaccines-09-01167-f002:**
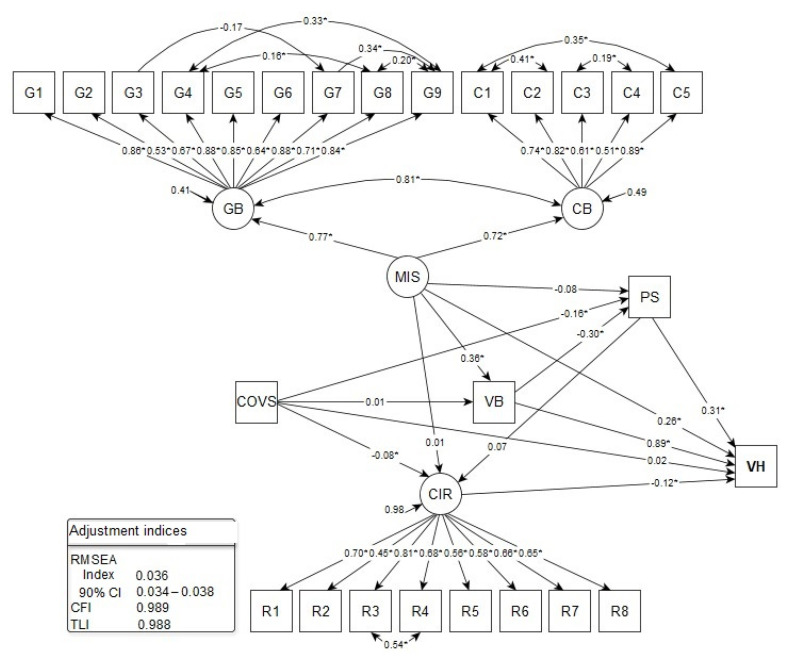
Structural equation model with specific direct and indirect effects for COVID-19 Vaccine hesitancy in populations of Portuguese-speaking countries. 2021. (N = 6843). CIR: COVID-19 Individual Responses (R1–R8: indicator variables); CB: Conspiracy Beliefs (C1–C5: indicator variables); GB: General Beliefs (G1–G9: indicator variables); MIS: COVID-19 Misinformation; COVS: COVID-19 Suffering; VB: Vaccine Conspiracy Beliefs; PS: Perceived Stress; VH: Vaccine Hesitancy. RMSEA: root mean square error of approximation; 90%CI: 90% confidence interval; CFI: comparative fit index; TLI: Tucker-Lewis Index. * *p*-value < 0.05.

**Table 1 vaccines-09-01167-t001:** Standardized factor loadings of individual response measurement models and COVID-19 misinformation in Portuguese-speaking countries. 2021. (N = 6843).

Latent Variables	Indicator Variables (Codes)	λ	*p*-Value
**CIR**			
	Avoiding bars, restaurants, and events (R1)	0.654	<0.001
	Using disinfectants for cleaning the environment (R2)	0.511	<0.001
	Hand hygiene with soap and water, and alcoholic solution (R3)	0.777	<0.001
	Postponing national or international travels (R4)	0.654	<0.001
	Working remotely (R5)	0.609	<0.001
	Supplying goods (R6)	0.614	<0.001
	Social isolation to prevent crowding (R7)	0.645	<0.001
	Using a face shield (R8)	0.609	<0.001
**CB**			
	The virus was created in the laboratory by Chinese scientists (C1)	0.770	<0.001
	There was genetic manipulation of the virus to cause AIDS (C2)	0.602	<0.001
	Social isolation can reduce immunity and facilitate virus infection (C3)	0.458	<0.001
	Positive asymptomatic people do not transmit the virus to other people (C4)	0.871	<0.001
	The virus was spread by the pharmaceutical industry for population control (C5)	0.722	<0.001
**GB**			
	Avocado, hibiscus, perfume aerosols, and whiskey tea have a preventive potential (G1)	0.851	<0.001
	Alcoholic solution is more efficient than washing hands with soap and water as a preventive measure (G2)	0.496	<0.001
	Daily use of alcohol gel can be toxic and extremely harmful to health (G3)	0.661	<0.001
	The virus can be eliminated from the body by drinking water and gargling with warm water, saline, or acidic solutions, thus preventing the infection (G4)	0.719	<0.001
	Vinegar is better than alcohol to avoid contamination by COVID-19 (G5)	0.637	<0.001
	Autohemotherapy is very effective against the new coronavirus (G6)	0.896	<0.001
	Eating garlic prevents contagion by the new coronavirus (G7)	0.881	<0.001
	The virus does not survive temperatures above 26 degrees (G8)	0.856	<0.001
	Drinking clean water every 15 min expels the new coronavirus, as it prevents it from going to the lungs (G9)	0.855	<0.001
r	CB1↔GB		
		0.949	<0.001
	CB ^a^		
	GB ^a^	0.559	<0.001
		0.796	<0.001
r ^b^	C11↔C2		
	C11↔C5	0.477	<0.001
	C31↔C4	0.385	<0.001
	G41↔G9	0.210	<0.001
	G71↔G9	0.316	<0.001
	CB1↔GB	0.217	<0.001

λ: Standardized factorial loads; CIR: COVID-19 individual responses; CB: Conspiracy beliefs; GB: General beliefs; MIS: COVID-19 Misinformation. ^a^ 1st order factor. ^b^ Residual correlations (↔) between indicators.

**Table 2 vaccines-09-01167-t002:** Standardized Regression Coefficients of structural equation models stratified by gender, age, and education, using VH as the response variable in populations of Portuguese-speaking countries (N = 6843).

Adjustment Pathways/Indices	Gender		Age (Years)			Education	
	Men	Women	18 to 29	30 to 49	50 or More	Elementary and High School	University
**Ancestors of VH**							
COVS→VH	−0.007	0.027	0.098	0.004	/0.040	0.004	0.001
MIS→VH	0.317 **	0.269 **	0.285 *	0.673 **	−0.008	0.233 **	0.245 **
CIR→VH	−0.152 *	−0.122 *	−0.291 *	−0.110	−0.410 **	−0.106 *	−0.098 *
PS→VH	0.297 **	0.344 **	0.578 **	0.111	0.151 *	0.320 **	0.318 **
GB→VH	0.795 **	0.953 **	0.587 **	0.484 **	0.906 **	0.911 **	0.905 **
**Descendants of COVS**							
COVS→PS	−0.265 **	−0.062	−0.106	−0.195 **	−0.077	−0.128 *	−0.128 *
COVS→GB	−0.085	0.068 *	0.037	0.008	0.051	0.037	0.037
COVS→CIR	−0.180 **	0.042	0.133 *	−0.188 **	0.066	−0.081 *	−0.110 **
**Descendants of MIS**							
MIS→PS	−0.059	−0.132	−0.148	−0.304 *	−0.043	−0.062	−0.067
MIS→GB	0.372 **	0.365 **	0.316 **	0.617 **	0.382 **	0.323 **	0.333 **
MIS→CIR	0.002	−0.007	−0.131	0.199 *	−0.119	−0.043	−0.022
**PS descendants**							
PS→CIR	0.138 *	0.088	0.359 **	0.039	0.066	0.054	0.049
**Descendants of VB**							
GB→PS	−0.163 *	−0.461 **	−0.155 *	−0.097	−0.200	−0.324 **	−0.320 **
r ^a^							
CB1↔GB	0.820 **	0.828 **	0.845 **	0.942 **	0.739 **	0.815 **	0.822 **
R21↔R3	0.376 **	0.361	0.187	0.594 *	0.517	−	0.587 **
C11↔C2	0.376 **	0.430 **	0.538 **	0.471 **	0.236 **	0.405 **	0.408 **
C11↔C5	0.338 **	0.361 **	0.353 **	0.353 **	0.498 **	0.339 **	0.345 **
C31↔C4	0.178 **	0.188 **	0.226 **	0.170 **	0.218 **	0.168 **	0.167 **
G31↔G7	−0.148 **	−0.187 **	−0.129 *	−0.206 **	−0.211 *	−0.202 **	−0.179 **
G41↔G9	0.484 **	0.270 **	0.427 **	0.171 **	0.331 **	0.280 **	0.309 **
G41↔G8	0.134 *	0.164 **	0.090	0.203 **	−0.011	0.160 **	0.158 **
G71↔G9	0.490 **	0.261 **	0.332 *	0.258 **	0.313 **	0.368 **	0.335 **
G81↔G9	0.258 **	0.180	0.262 **	0.109 *	0.176 *	0.188 **	0.187 **
**Fit indices**							
RMSEA							
Index	0.055	0.036	0.041	0.039	0.060	0.038	0.037
90%CI	0.051–0.059	0.037–0.046	0.026–0.029	0.036–0.042	0.054–0.066	0.036–0.041	0.034–0.039
CFI	0.975	0.987	0.995	0.988	0.995	0.989	0.988
TLI	0.971	0.985	0.995	0.986	0.995	0.987	0.987

CIR: COVID-19 individual responses; MIS: COVID-19 misinformation; COVS: COVID-19 suffering; VB: Vaccine conspiracy beliefs; PS: Perceived stress; VH: Vaccine hesitancy. ^a^ Residual correlations (↔) between indicators. * *p*-value < 0.05. ** *p*-value < 0.001.

**Table 3 vaccines-09-01167-t003:** Standardized total and indirect effects of the structural equation model using VH as the response variable in populations of Portuguese-speaking countries. (N = 6843).

Pathways	General	Gender		Age Group (Years)			Education	
		Men	Women	18 to 29	30 to 49	50 or More	Elementary and High School	University
**Total effects**								
COVS→VH	−0.009	−0.117*	0.056	0.028	0.008	−0.031	0.002	0.002
MIS→VH	0.523 **	0.579 **	0.517 **	0.416 **	0.911 *	0.372 **	0.480 **	0.495 **
CIR→VH	−0.122 *	−0.152 *	−0.122 *	−0.291 *	−0.110	−0.410 **	−0.106 *	−0.098 *
PS→VH	0.305 **	0.276 **	0.334 **	0.474 **	0.107	0.124	0.314 **	0.314 **
GB→VH	0.795 **	0.750 **	0.799 **	0.514 **	0.474 **	0.881 **	0.809 **	0.804 *
**Specific indirect effects**								
COVS								
COVS→PS→VH	−0.050 **	−0.079 **	−0.021	−0.061	−0.022	−0.012	−0.041 *	−0.041 *
COVS→GB→VH	0.008	−0.067	0.065 *	0.022	0.004	0.046	0.034	0.034
COVS→CIR→VH	0.010 *	0.027 *	−0.005	−0.039	0.021	−0.027	0.009 *	0.011 *
COVS→GB→PS→VH	−0.001	0.004	−0.011 *	−0.003	0.000	−0.002	−0.004	−0.004
COVS→PS→CIR→VH	0.001	0.006	0.001	0.011	0.001	0.002	0.001	0.001
COVS→GB→PS→CIR→VH	0.000	0.000	0.000	0.001	0.000	0.000	0.000	0.000
**MIS**								
MIS→PS→VH	−0.026	−0.018	−0.046	−0.085	−0.034	−0.006	−0.020	−0.021
MIS→GB→VH	0.322 **	0.296 **	0.348 **	0.185 **	0.299 **	0.346 **	0.295 **	0.301 **
MIS→CIR→VH	−0.001	0.000	0.001	0.038	−0.022	0.049	0.005	0.002
MIS→GB→PS→VH	−0.034 **	−0.018 *	−0.058 **	−0.028 *	−0.007	−0.012	−0.034 **	−0.034 **
MIS→PS→CIR→VH	0.001	0.001	0.001	0.015	0.001	0.001	0.000	0.000
MIS→GB→PS→CIR→VH	0.001	0.001	0.002	0.005	0.000	0.002	0.001	0.001
**PS**								
PS→CIR→VH	−0.008	−0.021	−0.011	−0.104 *	−0.004	−0.027	−0.006	−0.005
**GB**								
GB→PS→VH	−0.094 **	−0.048 *	−0.159 **	−0.089 *	−0.011	−0.030	−0.104 **	−0.102 *
GB→PS→CIR→VH	0.002	0.003	0.005	0.016	0.000	0.005	0.002	0.002

CIR: COVID-19 individual responses; MIS: COVID-19 misinformation; COVS: COVID-19 suffering; VB: Vaccine conspiracy beliefs; PS: Perceived stress; VH: Vaccine hesitancy. * *p*-value<0.05. ** *p*-value<0.001.

## Data Availability

Data connected to this research are available from the corresponding author under request (AFLS).

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
