# Peer review of "Determinants of COVID-19 Vaccine Hesitancy in Portuguese-Speaking Countries: A Structural Equations Modeling Approach"

_vaccines, 2021, doi:10.3390/vaccines9101167_

Round 1

Reviewer 1 Report

The manuscript titled “Determinants of COVID-19 vaccine hesitancy in Portuguese-speaking countries: a structural equations modeling approach” proposed a series of SEM models in order to estimate the prevalence and the factors associated with Covid-19 Vaccine Hesitancy (VH) in Portuguese speaking countries. A large sample of respondents was recruited through an online questionnaire. SEM methodology was used to analyze data. Results showed that the prevalence of VH was modulated by vaccine-related conspiracy beliefs, perceived stress, Covid-19 Misinformation, and individual responses to Covid-19. These effects, as well as indirect and total effects for different strata of gender, age, education level showed some fluctuations in their magnitude. Authors discussed their results in light of previous literature highlighting strengths and limitations of their work, suggesting also hints for further research.

I carefully read the manuscript, and I think it may be of interest for the readers of Vaccines. The manuscript is very well-written and properly addresses the interesting issue of the factors associated with vaccine hesitancy. The topic is really relevant nowadays, and more knowledge coming from well-conducted primary studies is needed. I also appreciated the construction of a well-structured model in order to provide a complete representation of the phenomenon as well as the differential analyses for each stratum of socio-demographic variables. I found that the introduction section and the methodology employed are clear and detailed, as well as the explanations provided in the discussion section. I only have few minor remarks:

Abstract Section

Page 2 line 66-68: the sentence is too verbose and unclear, it is difficult to understand the message. Please, try to better explain the concept.

Materials and Methods Section

Page 3 line 109, Sample subsection: what do the terms “incidence rate” and “accuracy” refer to? What formula and/or software did you use to estimate the sample size? Please add specifications to this procedure in order to make it replicable.

Page 3, lines 124-127: Please, can you provide the number of participants for each reason of exclusion?

Page 4 lines 164-166, Conceptual structure and study hypotheses subsection: you wrote “The latent constructs and other covariates that made up the structure of the theoretical model are defined below.” What do you mean by “covariates”? In a SEM model, each variable is meant to be covariated for all the other variables.

Results Section

Page 6, Prevalence of vaccine hesitancy section: the values of prevalence you reported are estimates of population prevalence measured on a sample. I think it would be useful to calculate and report 95% confidence interval for each estimate.

Page 8, Structural equation model of SARS-CoV-2 vaccine hesitancy section: in the previous paragraphs, Authors use the symbol “l” (Greek letter lambda) to identify the parameter “factor loading”. In the present paragraph, the same symbol is used to report the effects within the structural model. To the best of my knowledge, in a structural model one can encounter regression or correlation coefficients, but no more factor loadings since the latter pertain to the measurement model. Please, can you provide an explanation for the use of lambda in the structural model and throughout the whole manuscript?

Moreover, also in Table 1 the lambda symbol is attributed also to the residual correlations between indicators.

Page 10, Stratified structural equation models of SARS-CoV-2 vaccine hesitancy subsection, line 358: from this point on, results for models of different strata of socio-demographic variables are reported. Please, can you report the criteria for defining the different strata for each variable in the method section? For example, why the sample was divided in three age categories, and why those categories are 18-29, 30-49, and so on?

Author Response

1. Abstract Section

line 66-68: the sentence is too verbose and unclear, it is difficult to understand the message. Please try to better explain the concept.
- Thank you very much for this recommendation. We modified the passage to make it clearer.

2. Materials and Methods Section

Page 3 line 109, Sample subsection: what do the terms “incidence rate” and “accuracy” refer to? What formula and/or software did you use to estimate the sample size? Please add specifications to this procedure in order to make it replicable.
- All this information has been entered:
"The sample size was calculated in the G Power software (version 3.1.9.7), considering the total population size of the countries of interest (N= 286,165,991), an incidence rate of the studied phenomenon of 50% (because there are no previous studies with this population); an tolerable error of 3%, a sample design effect correction of 2, a 95% confidence level, and an additional number of 20% participants to compensate losses and refusals. The minimum sample size was calculated at 2,562 participants".

3. Page 3, lines 124-127: Please, can you provide the number of participants for each reason of exclusion?
- We corrected according to the suggestion of the reviewer "A total of 7,083 people answered the questionnaire. Portuguese speakers residing in other countries (n=127), under 18 years (n=21), and who failed to complete more than 50% of the mandatory questions (92) were considered ineligible. Thus, the final sample consisted of 6,843 participants".

4. Page 4 lines 164-166, Conceptual structure and study hypotheses subsection: you wrote “The latent constructs and other covariates that made up the structure of the theoretical model are defined below.” What do you mean by “covariates”? In a SEM model, each variable is meant to be covariated for all the other variables.
- The reviewer is correct. it was a bug we fixed, probably derived from the translation.

5. Page 6, Prevalence of vaccine hesitancy section: I think it would be useful to calculate and report 95% confidence interval for each estimate.
- We add 95%CI for each variable presented. We also add this information in the supplemental material.

6. Page 8, Structural equation model of SARS-CoV-2 vaccine hesitancy section: in the previous paragraphs, Authors use the symbol “l” (Greek letter lambda) to identify the parameter “factor loading”. In the present paragraph, the same symbol is used to report the effects within the structural model. To the best of my knowledge, in a structural model, one can encounter regression or correlation coefficients, but no more factor loadings since the latter valid to the measurement model. Please, can you provide an explanation for the use of lambda in the structural model and throughout the whole manuscript?
- The reviewer is right. In fact, the lambda represents the factor loading in the measurement model and the beta is the regression or correlation coefficients in the structural model. we made changes to the description of results and titles in figure 2, table 2 and table 3.

7. Moreover, also in Table 1 the lambda symbol is also attributed to the residual correlations between indicators.
- In this comment here there was a mistake by the reviewer. In table 1, the residual correlations are represented by the letter "r", as well as in the description of the main findings: "The high correlation between the CB and GB constructs (r=0.949) endorsed the convergence of the second-order factor MIS."

8. Page 10, Stratified structural equation models of SARS-CoV-2 vaccine hesitancy subsection, line 358: from this point on, results for models of different strata of socio-demographic variables are reported. Please, can you report the criteria for defining the different strategies for each variable in the method section? For example, why the sample was divided into three age categories, and why those categories are 18-29, 30-49, and so on?
-We corrected as indicated by the reviewer "Age strata were defined to enable the comparison between groups of younger, mid-dle-aged and older adults considering the age of the sample studied. The variable level of education was dichotomized into Elementary/High School (9 years of age or less corresponding to less education) and university education (higher education).

We hope the reviewer is satisfied with our effort to respond to the suggestions.
We are available for further clarification

Reviewer 2 Report

This is a very interesting study. The overall conclusions seem solid. However, this would be a difficult paper to read for the average reader. There are a large number of abbreviations, some that appear unnecessary. For example, there are some in the data analysis section that are used less the 3 times in the paper, and it just adds to complexity of the text. There appears to be a few "jargon" words or phrases that also add to the difficulty in reading. For example, the phrase "standardized factor loadings" was used, and although this may be clear to a selective group of scientist, it does make it difficult for someone not familiar with this terminology. So overall, this is a nice study with important information, but it would benefit from a revision that targets a more general audience.

Author Response

1. This is a very interesting study. The overall conclusions seem solid. However, this would be a difficult paper to read for the average reader. There are a large number of abbreviations, some that appear unnecessary. For example, there are some in the data analysis section that are used less than 3 times in the paper, and it just adds to the complexity of the text. There appears to be a few "jargon" words or phrases that also add to the difficulty in reading. For example, the phrase "standardized factor loadings" was used, and although this may be clear to a selective group of scientist, it does make it difficult for someone not familiar with this terminology. So overall, this is a nice study with important information, but it would benefit from a revision that targets a more general audience.

- Unfortunately, the method applied here is very specific and full of details. The use of acronyms is essential for the text not to become tiring. However, we tried to reduce the number of acronyms, but changing some names was not possible.

We hope the reviewer is satisfied with our effort to respond to the suggestions.
We are available for further clarification